# On-Demand Sampling:
# Learning Optimally from Multiple Distributions [*]

Nika Haghtalab, Michael I. Jordan, and Eric Zhao

University of California, Berkeley

## Abstract

Societal and real-world considerations such as robustness, fairness, social welfare and multi-agent tradeoffs have given rise to multi-distribution learning paradigms, such as *collaborative* [5], *group distributionally robust* [36], and *fair federated* learning [27]. In each of these settings, a learner seeks to minimize its worst-case loss over a set of $n$ predefined distributions, while using as few samples as possible. In this paper, we establish the optimal sample complexity of these learning paradigms and give algorithms that meet this sample complexity. Importantly, our sample complexity bounds exceed that of the sample complexity of learning a single distribution only by an additive factor of $\frac{n \log(n)}{\varepsilon^2}$. These improve upon the best known sample complexity of agnostic federated learning by Mohri et al. [27] by a multiplicative factor of $n$, the sample complexity of collaborative learning by Nguyen and Zakynthinou [29] by a multiplicative factor $\frac{\log n}{\varepsilon^3}$, and give the first sample complexity bounds for the *group DRO* objective of Sagawa et al. [36]. To achieve optimal sample complexity, our algorithms learn to sample and learn from distributions on demand. Our algorithm design and analysis extends stochastic optimization techniques to solve zero-sum games in a new stochastic setting.

## 1 Introduction

Pervasive needs for robustness, fairness, and multi-agent collaboration in learning have given rise to multi-distribution learning paradigms (e.g., [5, 36, 27, 12]). In these settings, we seek to learn a model that performs well on *any distribution* in a pre-defined set of interest. For fairness considerations, these distributions may represent heterogeneous populations of different protected or socio-economic attributes; in robustness applications, they may capture a learner's uncertainty regarding the true underlying task; and in muti-agent collaborative or federated applications, they may represent agent-specific learning tasks. In these applications, the performance and optimality of a model is measured by its worst test-time performance on a distribution in the set. We are concerned with this fundamental problem of designing sample-efficient multi-distribution learning algorithms.

The sample complexity of multi-distribution learning differs from that of learning a single distribution in several ways. On one hand, learning tasks of varying difficulty require different numbers of samples. On the other hand, similarity or overlap among learning tasks may obviate the need to sample from some distributions. This makes the use of a fixed per-distribution sample budget highly inefficient and suggests that optimal multi-distribution learning algorithms should *sample on demand*. That is, algorithms should take additional samples *whenever they need them* and *from whichever distribution* they want them. On-demand sampling is especially appropriate when some population data is scarce (as in fairness mechanisms in which samples are amended [32]); when the designer can actively

---

[*]Authors are ordered alphabetically. Addresses: `nika@berkeley.edu`, `jordan@cs.berkeley.edu`, `eric.zh@berkeley.edu`.

36th Conference on Neural Information Processing Systems (NeurIPS 2022).

| Problem | Sample Complexity | Thm | Best Previous Result |
|---|---|---|---|
| Collab. Learning UB | $\varepsilon^{-2}\left(\log|\mathcal{H}| + n\log(\frac{n}{\delta})\right)$ | [4.1] | $\varepsilon^{-5}\log\left(\frac{1}{\varepsilon}\right)\log(\frac{n}{\delta})(\log|\mathcal{H}| + n)$ [29] |
| Collab. Learning LB | $\varepsilon^{-2}(\log|\mathcal{H}| + n\log(\frac{k}{\delta}))$ | [4.2] | $\varepsilon^{-1}n\log(k/\delta)$ [5] |
| GDRO/AFL UB | $\varepsilon^{-2}\left(\log|\mathcal{H}| + n\log(\frac{n}{\delta})\right)$ | [4.1] | $\varepsilon^{-2}\left(n\log|\mathcal{H}| + n\log(\frac{n}{\delta})\right)$ [27] |
| GDRO/AFL UB | $\varepsilon^{-2}\left(D_{\mathcal{H}} + n\log(\frac{n}{\delta})\right)$ | [5.1] | N/A |
| (Training error convg.) | $\varepsilon^{-2}\left(D_{\mathcal{H}} + n\log(\frac{n}{\delta})\right)$ | [5.2] | $\varepsilon^{-2}D_{\mathcal{H}}$ (expected convergence only) [36] |

Table 1: This table gives upper (*UB*) and lower bounds (*LB*) on the sample complexity of learning model class $H$ on $n$ distributions. For the collaborative learning and AFL settings, the sample complexity upper bounds refer to the problem of learning a randomized model of worst-case error $\mathrm{OPT} + \varepsilon$ or a deterministic classifier of worst-case error $2\mathrm{OPT} + \varepsilon$. For the GDRO setting, sample complexity refers to learning a deterministic model with worst-case error of $\mathrm{R\text{-}OPT} + \varepsilon$, where R-OPT is the best worst-case error attainable in a convex compact model space $H$. $D_{\mathcal{H}}$ denotes the Bregman radius of $H$, and $k = \min\{n, \log|\mathcal{H}|\}$. Sample complexity bounds of Collaborative and Agnostic federated learning in existing works, extend to VC dimension and Rademacher complexity. Our results also extend to VC dimension under some assumptions.

perturb datasets towards rare or atypical instances (such as in robustness applications [21, 44]); or when sample sets represent agents' contributions to an interactive multi-agent system [27, 6].

Blum et al. [5] demonstrated the benefit of on-demand sampling in the *collaborative learning* setting, where all data distributions are realizable with respect to the same target classifier. This line of work established that learning $n$ distributions on-demand takes $\widetilde{O}\left(\log(n)\right)$ times the sample complexity of learning a single realizable distribution [5, 8, 29], whereas relying on batched uniform convergence takes $\widetilde{\Omega}\left(n\right)$ times that of learning a single distribution [5]. However, beyond the realizable setting, the best known multi-distribution learning results fall short of this promise: existing on-demand sample complexity bounds for agnostic collaborative learning have highly suboptimal dependence on $\varepsilon$, requiring $\widetilde{O}\left(\log(n)/\varepsilon^3\right)$ times the sample complexity of agnostically learning a single distribution [29]. On the other hand, agnostic federated learning bounds [27] have been studied only for algorithms that sample in one large batch and thus require $\widetilde{\Omega}\left(n\right)$ times the sample complexity of a single learning task. Moreover, the test-time performance of some key multi-distribution methods, such as group distributionally robust optimization [36], have not been studied from a provable or mathematical perspective before.

In this paper, we give a general framework for obtaining *optimal and on-demand sample complexity* for three multi-distribution learning settings. Table 1 summarizes our results. All three settings consider a set $\mathcal{D}$ of $n$ distributions and a model class $\mathcal{H}$. They evaluate the performance of a model $h$ (or a distribution over models) by its worst-case performance, $\max_{D \in \mathcal{D}} \mathrm{Risk}_D(h)$. As a benchmark, they consider the worst-case loss of the best model, i.e., $\mathrm{OPT} = \min_{h^* \in \mathcal{H}} \max_{D \in \mathcal{D}} \mathrm{Risk}_D(h^*)$. Importantly, all of our sample complexity upper bounds demonstrate only an *additive increase of* $\varepsilon^{-2}n\log(n/\delta)$ *over the sample complexity of a single learning task*, compared to the multiplicative factor increase required by existing works.

- *Collaborative learning of Blum et al. [5]:* For agnostic collaborative learning, our Theorem 4.1 gives a randomized and a deterministic model that achieve performance guarantees of $\mathrm{OPT} + \varepsilon$ and $2\mathrm{OPT} + \varepsilon$, respectively. Our algorithms have an optimal sample complexity of $O\left(\frac{1}{\varepsilon^2}(\log(|H|) + n\log(\frac{n}{\delta}))\right)$. This improves upon the work of Nguyen and Zakynthinou [29] in two ways. First, it provides error bounds of $\mathrm{OPT} + \varepsilon$ for randomized classifiers, where only $2\mathrm{OPT} + \varepsilon$ was previously established. Second, it improves the upper bound of Nguyen and Zakynthinou [29] by a multiplicative factor of $\log(n)/\varepsilon^3$. In Theorem 4.2, we give a matching lower bound on this sample complexity, thereby establishing the optimality of our algorithms.

- *Group distributionally robust learning (group DRO) of Sagawa et al. [36]:* For group DRO, we consider a convex and compact model space $\mathcal{H}$. Our Theorem 5.1 studies a model that achieves an $\mathrm{OPT} + \varepsilon$ guarantee on the worst-case test-time performance of the model with an on-demand sample complexity of $\mathcal{O}\left(\frac{1}{\varepsilon^2}(D_H + n\log(\frac{n}{\delta}))\right)$. Our results also imply a high-probability bound

for the convergence of group DRO *training error* that improves upon the (expected) convergence guarantees of Sagawa et al. [36] by a factor of $n$.

- *Agnostic federated learning of [27]:* For agnostic federated learning, we consider a finite class of hypotheses. Our Theorems 4.1 and 5.1 show that on-demand sampling can accelerate the generalization of agnostic federated learning by a factor of $n$ compared to batch results established by Mohri et al. [27]. Our results also imply matching high-probability bounds to Mohri et al. [27] on the convergence of the training error in the batched setting.

To achieve these results, we contribute new insights and techniques for solving stochastic zero-sum games with sources of randomization that differ in both cost and quality. We frame the multi-distribution learning problems as a stochastic zero-sum game with uncertain payoffs and utilize stochastic mirror descent and a variational perspective to solve the game. In this case, the maximizing player can be interpreted as a weight vector for distributions $\mathcal{D}$, specifying from which distributions future on-demand samples should be taken. These on-demand samples form a stochastic gradient for the players. However, the quality of these estimators, the number of samples needed for them, and whether they can be reused later on, differs between the two players. We extend the Stochastic Mirror Descent framework to optimally trade off these asymmetric needs for samples. In Section 3 we give an overview of this approach and its technical challenges and contributions.

## 1.1 Related Work

**Learning models.** There are many lines of work that study multi-distribution learning but which have evolved independently in separate communities. The field of *collaborative learning* concerns the learning of a shared machine learning model by multiple *stakeholders* that each desire a model with low error on their own data distribution. The line of work initiated by Blum et al. [5] studies on-demand sample complexity bounds for realizable collaborative learning and was later extended to several related settings (e.g., [29, 8, 7, 30]). The agnostic federated learning framework of Mohri et al. [27] poses an equivalent of the multi-distribution learning objective as a fair and intuitive target for federated learning algorithms, and studies it in the offline setting with data-dependent analysis. Multi-distribution learning also arises in distributionally robust optimization [4] as the Group DRO problem [17], which is is motivated by deep learning applications with multiple deployment domains or protected demographics. These works focus on an empirical perspective, but have discussed training error convergence in offline settings [17, 36, 37]. Multi-distribution learning is also related to a line of work on multi-source domain adaptation (e.g., [3, 24]) and multi-group fairness notions (e.g., [35, 38, 13]). We describe these parallel threads in more detail in Section A.

**Stochastic game equilibria.** Our approach relates to a line of research on using online algorithms to find min-max equilibria by playing no-regret algorithms against one another [34, 15, 31, 9, 10]. Online mirror descent (OMD) is one well-studied family of methods that can find approximate minima of convex functions, and also approximate min-max equilibria of convex-concave games, with high probability using noisy first-order information [33, 28, 16, 2]. We bring these online learning tools to bear on the problem of finding saddle points in robust optimization formulations. The primary technical difference between multi-distribution learning and traditional saddle-point optimization problems is that we have sample access to distributions instead of noisy local gradients.

## 2 Preliminaries

Let $\mathcal{X}$ be an instance space, $\mathcal{Y}$ a label space, and $\mathcal{Z} = \mathcal{X} \times \mathcal{Y}$ a space of datapoints. A data distribution $D$ is a joint probability distribution over $\mathcal{Z}$. We consider a hypothesis class $\mathcal{H}$ of a subset of functions mapping $\mathcal{X}$ to $\mathcal{Y}$. With each distribution $D$, define a loss function $\ell_D : \mathcal{H} \times \mathcal{Z} \to [0, 1]$ measuring the loss of hypothesis $h$ on data point $z \in \mathcal{Z}$. We write $\ell_D$ as $\ell$ when $D$ is clear from context. We denote the expected loss, i.e. risk, of a hypothesis $h \in \mathcal{H}$ under a data distribution $D \in \mathcal{D}$ by:

$$\mathrm{Risk}_D(h) \coloneqq \mathop{\mathbb{E}}_{(x,y) \sim D} \left[ \ell_D \left( h, (x, y) \right) \right].$$

Importantly, we only assume that $\ell_D$'s are bounded and make no other assumptions on losses or distributions. For a distribution over the hypothesis class, $p \in \Delta \mathcal{H}$, and a distribution over data distributions, $q \in \Delta \mathcal{D}$, we refer to their expected loss by $\mathrm{Risk}_q(p) \coloneqq \mathbb{E}_{D \sim q} \left[ \mathbb{E}_{h \sim p} \left[ \mathrm{Risk}_D(h) \right] \right]$.

**Collaborative Learning.** We will use the *collaborative PAC learning model* of Blum et al. [5] and its agnostic extensions by Nguyen and Zakynthinou [29]. In this setting, the goal is to guarantee small risk for *every* distribution in a collection. Formally, given a set of data distributions $\mathcal{D} := \{D_1, \ldots, D_n\}$, the goal of the learner is to learn a hypothesis $h$ such that, with probability $1 - \delta$,

$$\max_{D \in \mathcal{D}} \mathrm{Risk}_D(h) \leq \mathrm{OPT} + \varepsilon, \text{ where } \mathrm{OPT} := \min_{h \in \mathcal{H}} \max_{D \in \mathcal{D}} \mathrm{Risk}_D(h). \tag{1}$$

**Group Distribution Robustness.** We will also study the closely related setting of *group distributionally robust optimization (Group DRO)* of Sagawa et al. [36]. Formally, the group DRO setting considers a model set $\Theta$ that is a convex compact subset of the Euclidean space and a convex loss function $\ell : \Theta \times \mathcal{Z} \to [0, 1]$ that is assumed to be differentiable over $\Theta$. Given a set of data distributions $\mathcal{D} := \{D_1, \ldots, D_n\}$, the learner seeks a model $\theta \in \Theta$, such that, with probability $1 - \delta$,

$$\max_{D \in \mathcal{D}} \mathbb{E}_{(x,y) \sim D} [\ell(\theta, (x, y))] \leq \mathrm{R\text{-}OPT} + \varepsilon, \text{ where } \mathrm{R\text{-}OPT} := \min_{\theta \in \Theta} \max_{D \in \mathcal{D}} \mathbb{E}_{(x,y) \sim D} [\ell(\theta, (x, y))]. \tag{2}$$

There is a close relationship between the Group DRO setting and collaborative learning. In particular, when $\Theta = \Delta(\mathcal{H})$ and $\mathcal{H}$ is finite, the two goals are analogous but with two exceptions: first, the Group DRO could return a distribution over functions while collaborative learning requires the solution to be a deterministic function, and second, R-OPT is potentially more competitive than OPT since it allows randomization. We note that the group DRO setting is equivalent to the agnostic federated learning framework of [27], thus our results for DRO extend to that setting as well.

**Sample complexity.** We are interested in the design of algorithms that achieve the above goals while using a small number of samples from distributions $D_1, \ldots, D_n$. We formalize the sample complexity by the total number of calls made to *example oracles* $\mathrm{EX}(D_i)$. Each call $\mathrm{EX}(D)$ produces an i.i.d. sample from $D$. We note that these example oracles also allow us to sample from any mixture distribution $q \in \Delta \mathcal{D}$, e.g., by first selecting a $D_i$ according to the mixture and then calling $\mathrm{EX}(D_i)$.

## 2.1 Technical Background

We will use tools and definitions from the literature on zero-sum games and no-regret learning throughout the paper. This section provides a brief overview of these concepts.

**Zero-Sum Games.** A finite two-player zero-sum game is described by the tuple $(A_-, A_+, \phi)$ where $A_- = \{1, \ldots, n\}$ and $A_+ = \{1, \ldots, m\}$ are finite sets of actions and where $\phi : A_- \times A_+ \to [0, C]$. In this game, the players choose *mixed strategies* over actions sets. These are distributions that are denoted by a vector of probabilities $p \in \Delta A_-$ and $q \in \Delta A_+$. The expected payoff of mixed strategies is denoted by $\phi(p, q) = \mathbb{E}_{i \sim p, j \sim q} [\phi(i, j)]$. The goal of the minimizing player is to minimize this expected payoff and the maximizer seeks to maximize the expected payoff; that is, to solve

$$\min_{p \in \Delta A_-} \max_{q \in \Delta A_+} \phi(p, q).$$

A pair $(p, q)$ that solves this optimization problem is called a *min-max equilibrium*. Similarly, a solution is called an *$\varepsilon$-min-max* equilibrium if neither player can unilaterally improve their objective by more than $\varepsilon$. Formally, $(p, q)$ is an $\varepsilon$-min-max equilibrium if both players' regrets are at most $\varepsilon$, i.e., $\mathrm{Reg\text{-}Min}(p, q) := \phi(p, q) - \min_{i^* \in A_-} \phi(i^*, q) \leq \varepsilon$ and $\mathrm{Reg\text{-}Max}(p, q) := \max_{j^* \in A_+} \phi(p, j^*) - \phi(p, q) \leq \varepsilon$. We will next describe methods that find $\varepsilon$-min-max equilibria by finding solutions $(p, q)$ for which $\mathrm{Reg\text{-}Min}(p, q) + \mathrm{Reg\text{-}Max}(p, q)$ is at most $\varepsilon$. We describe a more general formulation for convex-concave zero-sum games in Appendix B.1 which we will use for the Group DRO problem.

**No-Regret Learning.** We consider an online setting where an arbitrary set of *operators*, $g^{(1)}, \ldots g^{(T)} \in \mathcal{E}^*$, is revealed sequentially to a learner who must choose a matching sequence of actions, $w^{(1)}, \ldots w^{(T)}$, from a convex compact set $Z \subseteq \mathcal{E}$. Here, $\mathcal{E}$ and $\mathcal{E}^*$ respectively refer to an arbitrary Euclidean space and its dual. We focus on a setting where an online learner commits to action $w^{(t)} \in Z$ before seeing $g^{(t)}, g^{(t+1)}, \ldots$ and aims to achieve vanishing *variational error* $\mathrm{Err}_\mathbf{V}(w^{(1:T)})$ defined by

$$\mathrm{Err}_\mathbf{V}(w^{(1:T)}) := \max_{w^* \in Z} \frac{1}{T} \sum_{t=1}^{T} \left\langle g^{(t)}, w^{(t)} - w^* \right\rangle. \tag{3}$$

We will denote no-regret algorithms by their update rule $\mathcal{Q} : \{Z \times \mathcal{E}^*\} \to Z$, where $\{Z \times \mathcal{E}^*\}$ denotes the space of arbitrary length sequences of action-operator pairs. Given a history sequence $w^{(1)}, \ldots, w^{(t)} \in Z$ and operator sequence $g^{(1)}, \ldots, g^{(t)} \in \mathcal{E}^*$, the algorithm returns $w^{(t+1)} = \mathcal{Q}\left(\{w^{(1)}, g^{(1)}\}, \ldots, \{w^{(t)}, g^{(t)}\}\right)$. When the history is clear from context, we write $w^{(t+1)} = \mathcal{Q}\left(w^{(t)}, g^{(t)}\right)$ as shorthand. For the particular case where $Z = \Delta^n$ is a probability simplex, one such algorithm is Exponential Gradient Descent (also known as Hedge):

$$\mathcal{Q}_{\text{hedge}}\left(\left\{w^{(1)}, g^{(1)}\right\}, \ldots, \left\{w^{(t)}, g^{(t)}\right\}\right) \coloneqq \frac{\widetilde{w}}{\|\widetilde{w}\|_1} \text{ where } \widetilde{w}_i \coloneqq w_i^{(t)} \exp\left\{-\eta g_i^{(t)}\right\}, \widetilde{w} \in \mathbb{R}^n \quad (4)$$

where $\eta$ is a user-defined step size, and $w_1$ is a user-defined initial iterate. By default, we take $w_1 = \left[\frac{1}{n}\right]^n$. The following lemma is a classical result on the variational error of exponential gradient descent.

**Lemma 2.1** ([40]). *Let $g^{(1)}, \ldots, g^{(T)} \in \mathbb{R}^n$ and $Z = \Delta^n$. Further assume $\left\|g^{(t)}\right\|_\infty \le C$ for all timesteps $t = 1, \ldots, T$. Choosing $\eta = \sqrt{\log n / T}$, after $T$ iterations of exponential gradient descent, the outputs $w^{(1)}, \ldots, w^{(T)}$ satisfies,*

$$\mathrm{Err}_{\mathbf{V}}(w^{(1:T)}) \le \frac{3C}{2}\sqrt{\frac{KL\left(w^{(T)}\|w^{(1)}\right)}{T}}.$$

## 3  Technical Overview of Our Approach

In this section, we provide an overview of our technical approach for addressing the sample complexity of collaborative learning and group DRO problems. In later sections, we will refer to the approach outlined in this section to sketch proofs and design algorithms. We will focus our exposition on collaborative learning and briefly indicate how the same approach applies to the group DRO setting.

At a high level, we first frame collaborative learning as a zero-sum game with uncertain payoffs and aim to use a variational perspective to learn its minmax equilibrium. We specifically choose the variational perspective (instead of an arbitrary online learning approach), since it allows us to linearize the effect of uncertain payoffs on the resulting error. We then use stochastic gradients to solve the variational problem. Our stochastic gradients will rely on i.i.d. samples from the distributions to estimate gradients both with respect to distributions over $\mathcal{H}$ and mixtures over $\mathcal{D}$ but with an asymmetric bound on the bias and variance of the estimates. Along the way, we develop tools and formalisms that handle the asymmetric cost of stochastic gradients and obtain optimal sample complexity results. We now address these steps in more detail.

**Collaborative Learning as Zero-Sum Games.**  When the hypothesis class $\mathcal{H}$ is finite, the collaborative learning problem with distribution set $\mathcal{D}$ corresponds to a zero-sum game $(A_-, A_+, \phi)$ with $A_- = \mathcal{H}, A_+ = \mathcal{D}$, and $\phi(i, j) = \mathrm{Risk}_j(i)$, where $i \in A_-$ and $j \in A_+$. Observe that the value of the min-max solution is equivalent to R-OPT. It is not hard to see that any $\varepsilon$-min-max equilibrium $(p, q)$ of this game corresponds to a $2\varepsilon$ collaborative learning solution, i.e.,

$$\mathbb{E}_{h \sim p}\left[\max_{D \in \mathcal{D}} \mathrm{Risk}_D(h)\right] \le \mathrm{OPT} + 2\varepsilon. \quad (5)$$

This enables us to use tools that have been developed for solving zero-sum games in order to address collaborative learning and group DRO settings. We will use a similar construction when hypothesis class $\mathcal{H}$ has finite VC dimension, where $A_-$ will instead refer to an appropriate $\varepsilon$-cover of $\mathcal{H}$.

**Using the Variational Error to deal with Payoff Uncertainty.**  A sufficient condition for minimizing regret, and thus finding $\varepsilon$-min-max equilibrium, is minimizing the variational error (Equation 3). In particular, for any finite zero-sum game $(A_-, A_+, \phi)$, defining $Z = [\Delta A_-, \Delta A_+]$ and operators

$$g^{(t)} = \left[\left\{\partial_{p_i}\phi(p^{(t)}, q^{(t)})\right\}_{i \in A_-}, \left\{-\partial_{q_j}\phi(p^{(t)}, q^{(t)})\right\}_{j \in A_+}\right], \quad (6)$$

ensures that variational error provides an upper bound on regret: $\mathrm{Err}_{\mathbf{V}}(w^{(1:T)}) \ge \mathrm{Reg\text{-}Min}(p, q) + \mathrm{Reg\text{-}Max}(p, q)$, where $w = (p, q)$ (see Fact C.1). In collaborative learning, when $p^{(t)}$ is the min-player's distribution over hypotheses and $q^{(t)}$ is max-player's distribution over the mixtures, the

gradient vectors refer to the risks of each hypothesis or distribution under $q^{(t)}$ or $p^{(t)}$ respectively:

$$g^{(t)} = [g_-^{(t)}, g_+^{(t)}], \quad g_-^{(t)} = \left\{ \mathrm{Risk}_{q^{(t)}}(h) \right\}_{h \in \mathcal{H}}, \quad g_+^{(t)} = \left\{ \mathrm{Risk}_D(p^{(t)}) \right\}_{D \in \mathcal{D}}. \qquad (7)$$

In the collaborative learning setting, we can only create noisy estimates $\widehat{g}$ for these gradients from samples. No-regret algorithms are advantageous in this setting as they choose their $t$th iterate $w^{(t)}$ before seeing the $t$th gradient $g^{(t)}$. This means that $w^{(t)}$ is independent of gradient noise, $\varepsilon^{(t)} := g^{(t)} - \widehat{g}^{(t)}$. We can thus linearize the noise and decompose variational error into the *training* and *generalization* errors as follows

$$\mathrm{Err}_{\mathbf{V}}(w^{(1:T)}) \leq \max_{w^* \in \Delta^n} \frac{1}{T} \sum_{t=1}^{T} \left\langle \widehat{g}^{(t)}, w^{(t)} - w^* \right\rangle + \max_{w^* \in \Delta^n} \frac{1}{T} \sum_{t=1}^{T} \left\langle \varepsilon^{(t)}, w^{(t)} - w^* \right\rangle. \qquad (8)$$

In contrast, generic no-regret algorithms that do not solve the variational inequality (e.g., when one player plays Hedge and another plays clairvoyant best-response as used in existing work in collaborative learning due to Blum et al. [5], Nguyen and Zakynthinou [29], Chen et al. [8]) nest the generalization and training errors which leads to a multiplicative increase in sample complexity.

**Leveraging Noisy Stochastic Gradients.** We will work with stochastic estimators of $g$. These are functions $\widehat{g} : \boldsymbol{\xi} \times \Delta A_- \times \Delta A_+$ of some external source of randomness, $\xi \in \boldsymbol{\xi}$, and a strategy profile of interest. For collaborative learning, the randomness source $\xi$ is an i.i.d.-sampled data point from an appropriate mixture of distributions and the estimator $\widehat{g}$ is then the empirical loss on this sample, which is an unbiased and bounded estimator in the range of the loss function, i.e., $[0, 1]$.

Interestingly, estimators of these stochastic gradients have an asymmetric need for data. As seen in Equation 7, the min-player's gradient $g_-(p, q)$ includes the risk of every hypothesis $h \in \mathcal{H}$ for the same data distribution $q$. Therefore, an unbiased estimator $\widehat{g}_-(p, q)$ can be constructed from a single call to an example oracle $\mathrm{EX}(q)$. We call this source of randomness $\xi^q$ and say that its cost is $r_- = 1$. While $\xi^q$ costs 1 unit, the randomness it provides is specialized to the point of inquiry, that is, it cannot be used for estimating other $\widehat{g}_-(p, q')$. We call this source of randomness and its associated unbiased estimation a *locally* unbiased estimator.

On the other hand, the max-player's gradient $g_+(p, q)$ includes the risk of the same hypothesis $p$ on *every distribution* $D \in \mathcal{D}$. Therefore, an unbiased estimator $\widehat{g}_+(p, q)$ *requires $n$ samples*, i.e., a call to every example oracle $\mathrm{EX}(D_i)$. We call this source of randomness producing $n$ samples $\xi^p$ and say that its cost is $r_+ = n$. Importantly, while $\xi^p$ costs $n$ unit, the randomness it provides can be reused for estimating other gradients, that is, it can provide an unbiased and bounded estimators for all $\widehat{g}_+(p', q')$. We call this source of randomness and its associated unbiased estimator a *globally* unbiased estimator. To emphasize the fact that this source of randomness is agnostic to $(p, q)$ we refer to it by $\xi^\perp$ hereafter. We refer the reader to Appendix B.2 for a more formal definition and description of these asymmetries.

**Minimizing Regret with Asymmetric Cost.** With the goal of minimizing sample complexity in mind, it is essential that we reuse randomness $\xi^\perp$ across $n$ time steps of variational algorithms. To do this, we introduce a stochastic variational approach in Algorithm 1 that accommodates different sampling frequencies for the minimizing and maximizing players. This will decouple the sample complexity of the minimizing agent (who requires a time horizon of at least $\log(A_-) \approx \log(\mathcal{H})$) and the maximizing agent. Lemma 3.1 proves this decoupling allows us to find an $\varepsilon$-min-max equilibrium with an additive $n + \log(\mathcal{H})$ sample complexity instead of a multiplicative $n \log(\mathcal{H})$.

Algorithm 1 uses the same randomness $\xi^{\perp(a)}$ of cost $r$ for estimating $g_+(p^t, q^t)$ for all $t \in [ar + 1, \ldots, a(r + 1)]$. On the other hand, the algorithm uses fresh randomness $\xi^{(t)}$ of cost 1 to estimate $g_-(p^t, q^t)$ for every time step $t$. The total randomness cost of this algorithm is thus $2T$ because iteration of the outer loop incurs $2r$ cost.

**Lemma 3.1.** *Let $(A_-, A_+, \phi)$ be a finite zero-sum game. Assume there exists $\xi^{q^{(t)}}$ of cost 1 providing locally unbiased estimates $\widehat{g}_-(\cdot)$ and there exists $\xi^{\perp(a)}$ of cost $r$ providing globally unbiased estimates $\widehat{g}_+(\cdot)$. With probability $1 - \delta$, Algorithm 1 returns an $\varepsilon$-min-max equilibrium of the game, so long as*

$$T \geq \frac{18}{\varepsilon^2} \left( \max \left\{ \frac{9 \log |A_-|}{4}, 8 \log \left( \frac{r+1}{\delta} \right) \right\} + \max \left\{ \frac{9 \log |A_+|}{4}, \frac{8r^2}{r+1} \log \left( \frac{r+1}{\delta} \right) \right\} \right). \qquad (9)$$

*Moreover, the total cost of randomness incurred by the algorithm is at most $2T$.*

---

**Algorithm 1** Finding Equilibria in Finite Zero-Sum Games with Asymmetric Costs.

---

**Output:** Mixed strategy profile $(p, q) \in \Delta A_- \times \Delta A_+$;
**Input:** Action sets $A_-, A_+$, cost $r \in \mathbb{Z}_+$, timesteps $T$, iterates $p^{(1)}, q^{(1)}$, gradient estimators $\widehat{g}_-, \widehat{g}_+$;
**for** $a = 1, 2, \ldots, \lceil T/r \rceil$ **do**
   Realize $\xi^{\perp^{(a)}}$ at cost $r$;      // Sample datapoints from every distribution.
   **for** $t = ar + 1 - r, \ldots, ar$ **do**
      Realize $\xi^{q^{(t)}}$ at cost $1$;      // Sample from adversary-selected distribution.
      Estimate gradients: $\widehat{g}_+^{(t)} = \widehat{g}_+\left(\xi^{\perp^{(a)}}, p^{(t)}, q^{(t)}\right), \quad \widehat{g}_-^{(t)} = \widehat{g}_-\left(\xi^{q^{(t)}}, p^{(t)}, q^{(t)}\right)$;
      Run Hedge updates: $p^{(t+1)} = \mathcal{Q}_{\text{hedge}}\left(p^{(t)}, \widehat{g}_+^{(t)}\right), q^{(t+1)} = \mathcal{Q}_{\text{hedge}}\left(q^{(t)}, \widehat{g}_+^{(t)}\right)$;
   **end for**
**end for**
Return the uniformly mixed strategies $\overline{p} = \frac{1}{T}\sum_{t=1}^{T} p^{(t)}$ and $\overline{q} = \frac{1}{T}\sum_{t=1}^{T} q^{(t)}$;

---

*Proof sketch.* Our approach uses Equation 8 to decompose the variational error into training error and generalization error. Since exponential gradient descent is known to bound the training error (as shown in Lemma C.4), it only remains to bound the generalization error (the second term in Equation 3). We note that in expectation each summand $\langle \varepsilon^{(t)}, w^{(t)} - w^* \rangle$ is zero. This is because $\varepsilon^{(t)} = g^{(t)} - \widehat{g}^{(t)}$ and $\widehat{g}^{(t)}$ are unbiased estimators. Therefore, the sum of these terms has an intuitive martingale interpretation and could be bounded by the Azuma-Hoeffding inequality.

There is a subtlety here, however. When we reuse the maximizing player's randomness over $r$ rounds, we create correlations between these terms in the generalization error that cannot be directly accommodated by a martingale. The trick here is to note that these correlations are entirely contained in $r$-length periods. So, we can partition our sequence to $r$ martingales and bound each one. This completes the proof. See Appendix C.1 for detailed proof of this lemma. □

**Derandomization.** The $\varepsilon$-min-max equilibria $(\overline{p}, \overline{q})$ returned by Exponentiated Gradient Descent gives a probability distribution $\overline{p}$ over the hypothesis class $\mathcal{H}$ that achieves the collaborative learning bound. To obtain a deterministic hypothesis, we can instead work with $h_p^{Maj}$ whose predictions are $p$-weighted majority votes over the hypotheses in $\mathcal{H}$. As stated below, the error of this deterministic classifier is approximately bounded by the expected error of $\overline{p}$.

**Lemma 3.2.** *For any $p \in \Delta\mathcal{H}$, $\max_{D \in \mathcal{D}} \text{Risk}_D(h_p^{Maj}) \leq 2\max_{D \in \mathcal{D}} \text{Risk}_D(p)$.*

This lemma in particular implies that for any $\varepsilon$-min-max equilibria $(\overline{p}, \overline{q})$, we have

$$\max_{D \in \mathcal{D}} \text{Risk}_D(h_{\overline{p}}^{Maj}) \leq 2\text{R-OPT} + 4\varepsilon \leq 2\text{OPT} + 4\varepsilon.$$

# 4 Collaborative Learning Bounds

In this section, we characterize the sample complexity of collaborative learning by providing tight upper and lower bounds for this problem. We describe Algorithm 2, which attains near-optimal sample complexity by on-demand sampling: iteratively selecting distributions to sample from.

## 4.1 Sample Complexity Upper Bounds

We are now prepared to describe our collaborative learning algorithm and guarantees, using the tools we developed in Section 3. Algorithm 2 is a direct application of Algorithm 1 to a zero-sum game with action sets $A_- = \mathcal{H}$, $A_+ = \mathcal{D}$ and payoff $\phi(h, D) = \text{Risk}_D(h)$. Here, $\xi^{q^{(t)}}$ makes one call to $\text{EX}(q^{(t)})$ and $\xi^{\perp(a)}$ makes one call to $\text{EX}(D)$ for each $D \in \mathcal{D}$. In other words, Algorithm 2 constructs distributions $p^{(t)} \in \Delta\mathcal{H}$ and $q^{(t)} \in \Delta\mathcal{D}$ by running the Hedge update. The gradient estimators used by Hedge are the empirical losses on a set of independent random variables. In particular, the minimizing player uses gradients $\ell_D(h, z^{(t)})$ for all $h \in \mathcal{H}$ for a single sample $z^{(t)} \sim \text{EX}(D)$ with $D \sim q^{(t)}$ and the maximizing player uses gradients $\ell_D(p^{(t)}, z_D^a)$ for all distributions $D \in \mathcal{D}$

---
**Algorithm 2** On-Demand Agnostic Collaborative Learning.
---
**Input:** Hypothesis class $\mathcal{H}$, distribution set $\mathcal{D}$ with $n := |\mathcal{D}|$;
**Initialize:** $p^{(1)} = [1/|\mathcal{H}|]^{|\mathcal{H}|}, q^{(1)} = [1/n]^n$, and iterations $T = \frac{36}{\varepsilon^2}(9\log(|\mathcal{H}|) + 35n\log(n/\delta))$;
**for** $a = 1, 2, \ldots, \lceil T/n \rceil$ **do**
    For all $D \in \mathcal{D}$, sample datapoint $z_D^a$ from EX$(D)$ .
    **for** $t = an + 1 - n, \ldots, an$ **do**
        Sample $z^{(t)}$ from EX$(D)$ with $D \sim q^{(t)}$ and estimate $\widehat{g}_-^{(t)} = [\ell_D(h, z^{(t)})]_{h \in \mathcal{H}}, \widehat{g}_+^{(t)} = [\ell_D(p^{(t)}, z_D^a)]_{D \in \mathcal{D}}$;
        Run Hedge updates: $p^{(t+1)} = \mathcal{Q}_{\text{hedge}}\left(p^{(t)}, \widehat{g}_-^{(t)}\right), q^{(t+1)} = \mathcal{Q}_{\text{hedge}}\left(q^{(t)}, \widehat{g}_+^{(t)}\right)$;
    **end for**
**end for**
**Return:** probability distribution over $\mathcal{H}$ given by the uniform mixture $\frac{1}{T}\sum_{t=1}^T p^{(t)}$.

---

where a single sample $z_D^a \sim$ EX$(D)$ is drawn per distribution and is reused for all time steps $t \in [(a-1)n + 1, \ldots, an]$.

Our main result in this section bounds the sample complexity of Algorithm 2.

**Theorem 4.1.** *For any finite hypothesis class $\mathcal{H}$ and unknown set of distributions $\mathcal{D}$, with probability $1 - \delta$, Algorithm 2 returns a distribution $\overline{p} \in \Delta\mathcal{H}$ such that*

$$\mathbb{E}_{h \sim \overline{p}}\left[\max_{D \in \mathcal{D}} \text{Risk}_D(h)\right] \leq OPT + \varepsilon \quad and \quad \max_{D \in \mathcal{D}} \text{Risk}_D(h_{\overline{p}}^{Maj}) \leq 2OPT + \varepsilon,$$

*using a number of samples that is $\mathcal{O}\left(\frac{\log|\mathcal{H}| + n\log(n/\delta)}{\varepsilon^2}\right)$.*

*Proof sketch.* By construction, Lemma 3.1 guarantees that with probability at least $1 - \delta$, the pair $(\overline{p}, \overline{q})$ is an $\varepsilon/2$-min-max equilibrium for the corresponding zero-sum game. As shown by Equation 5, $\overline{p}$ is a randomized classifier that meets the collaborative learning objective, i.e., its expected worst-case error is OPT $+ \varepsilon$. By Lemma 3.2, the corresponding deterministic classifier $h_{\overline{p}}^{Maj}$ has worst-case error of 2OPT $+ \varepsilon$. This bounds the error of the resulting classifier.

To bound the sample complexity, Lemma 3.1 shows that the randomness cost of Algorithm 1 is at most $2t$. Since the cost of randomness is exactly the total number of samples we take from our example oracles, the total sample complexity of Algorithm 2 is $2t \in \mathcal{O}\left(\varepsilon^{-2}\left(\log|\mathcal{H}| + n\log(n/\delta)\right)\right)$. $\qquad\square$

An analogue of Theorem 4.1 (Theorem C.3) holds for the case of infinite hypothesis classes of bounded Littlestone dimension with a sample complexity of $\mathcal{O}\left(\varepsilon^{-2}\left(\text{Little}(\mathcal{H}) + n\log(n/\delta)\right)\right)$. A similar result also holds with dependence on the VC dimension of $\mathcal{H}$ only (which is smaller than its Littlestone dimension) when additional assumptions hold. For example, if a hypothesis class $\mathcal{H}'$ is known in advance that is an $\varepsilon$-net of $\mathcal{H}$ with respect to every distribution in $\mathcal{D}$, one can instead run Algorithm 2 with a hypothesis class $\mathcal{H}'$. Such an $\varepsilon$-net of size $n\varepsilon^{-\mathcal{O}(\text{VCD}(\mathcal{H}))}$ necessarily exists; for example, the union of $\varepsilon$-nets with respect to each distribution $D \in \mathcal{D}$. It is also not strictly necessary to know an $\varepsilon$-net in advance. Instead, one can compute a net from samples or from other information about distributions in $\mathcal{D}$. In Appendix C.5, we explore a range of assumptions that allow us to compute such an $\varepsilon$-net from samples, without incurring a significant increase in the sample complexity of Theorem 4.1.

We end this subsection with a few remarks about our sample complexity upper bound.

**Remark 4.1.** *One question left open by these results is, for agnostic multi-distribution learning, whether it is possible to achieve sample complexity rates of $\mathcal{O}\left(\varepsilon^{-2}\left(\log(n)\text{VCD}(\mathcal{H}) + n\log(n/\delta)\right)\right)$ without any additional assumptions or a priori knowledge of an $\varepsilon$-net. It also remains open whether the $\log(n)$ factor in the $\log(n)\text{VCD}(\mathcal{H})/\varepsilon^2$ term is necessary for some VC classes, as Theorem 4.1 proves it is not necessary for some (e.g., finite) VC classes.*

**Remark 4.2.** *Theorem 4.1 improves over the best-known sample complexity for agnostic collaborative learning by Nguyen and Zakynthinou [29] in two ways, giving an $OPT + \varepsilon$ bound for randomized classifiers instead of their $2OPT + \varepsilon$ bound, and improving their sample complexity of $\mathcal{O}\left(\frac{1}{\varepsilon^5}\left(\log(n)\log(|\mathcal{H}|)\log\left(\frac{1}{\varepsilon}\right) + n\log\left(\frac{n}{\delta}\right)\right)\right)$ by a multiplicative factor of $\frac{1}{\varepsilon^3}\log(n)\log\left(\frac{1}{\varepsilon}\right)$.*

**Remark 4.3.** *For constants $\varepsilon$ and $\delta$, our sample complexity of $\mathcal{O}\left(\log(|\mathcal{H}|) + n \log n\right)$ appears to violate the lower bound of $\Omega\left(\log(|\mathcal{H}|) \log n + n \log \log |\mathcal{H}|\right)$ due to Chen, Zhang, and Zhou [8]. This discrepancy is due to a small error in the proof of that lower bound, which we have verified in private communications with the authors. In the next subsection, we give lower bounds on the sample complexity of collaborative learning that match our upper bounds.*

### 4.2 Sample Complexity Lower Bound

We now provide matching lower bounds for agnostic collaborative learning. Our lower bounds hold for collaborative learning algorithms obtaining error of R-OPT $+ \varepsilon$, using a randomized or deterministic hypothesis. We call an algorithm an $(\varepsilon, \delta)$-collaborative learning algorithm if for any collaborative instances it attains an error of R-OPT $+ \varepsilon$ with probability at least $1 - \delta$.

**Theorem 4.2.** *Take any $n, d \in Z_+$, $\varepsilon, \delta \in (0, 1/8)$, and $(\varepsilon, \delta)$-collaborative learning algorithm $A$. There exists a collaborative learning problem $(\mathcal{H}, \mathcal{D})$ with $|\mathcal{D}| = n$ and $|\mathcal{H}| = 2^d$, on which $A$ takes at least $\Omega\left(\frac{1}{\varepsilon^2}\left(\log |\mathcal{H}| + |\mathcal{D}| \log(\min\{|\mathcal{D}|, \log |\mathcal{H}|\}/\delta))\right)\right)$ samples.*

*Proof sketch.* We defer the formal proof of this theorem to Appendix C.3 and sketch the main ideas here. Let $\mathcal{X} = \{1, \ldots, d\}$, $\mathcal{Y} = \{+, -\}$, and $\mathcal{H}$ be the set of all functions $\mathcal{X} \to \mathcal{Y}$. Our construction combines two sets of hard distributions. Consider the case when $n = d \cdot \eta$ for some $\eta \in \mathbb{Z}$. First, we can reduce to a $d$-armed multi-arm bandit exploration problem giving us an $\Omega\left(d \log(1/\delta)/\varepsilon^2\right)$. Second, we construct $\eta$ hard instances on $\eta$ corresponding points. Since the learning algorithms has to solve each problem it has to incur a loss of $\eta \cdot d \log(d/\delta)/\varepsilon^2$. □

## 5 Group DRO and Agnostic Federated Learning

The results we describe in the collaborative learning setting can be generalized to the group DRO setting, and equivalently, agnostic federated learning.

**Theorem 5.1.** *Consider a group distributionally robust problem $(\Theta, \mathcal{D})$ with convex compact unit-diameter parameter space $\Theta$ of Bregman radius $D_\Theta$ (Definition B.11), and convex loss $\ell : \Theta \times \mathcal{Z} \to [0, C]$. A variant of Algorithm 2 (in particular Algorithm 4 in Appendix 4.1), returns $\overline{\theta} \in \Theta$ such that $\max_{D \in \mathcal{D}} \mathbb{E}_{z \sim D}\left[\ell(\theta, z)\right] \leq R\text{-}OPT + \varepsilon$, using a number of samples that is $\mathcal{O}\left(\frac{D_\Theta C^2 + nC^2 \log(n/\delta)}{\varepsilon^2}\right)$.*

The proof of this lemma is deferred to Appendix 4.1 and is similar to the proof of Theorem 4.1 except that it uses a generalization of Lemma 3.1 for general convex-concave games. This theorem establishes a generalization bound for the problem of group distributionally robust optimization [36] and improves, by a factor of $n$, existing sample complexity bounds for agnostic federated learning [27]. This improvement is attained by sampling data on-demand, whereas [27] only chooses a fixed distribution over groups/clients to sample from; this highlights the importance of adapting one's sampling strategy on-the-fly when learning robust models.

Another important question is how fast the training error of stochastic gradient descent converges for the group DRO/AFL settings and was considered by Sagawa et al. [36]. We can transfer our generalization guarantees for on-demand settings into batch settings and achieve the following corollary, which improves on the convergence guarantees of Sagawa et al. [36] by a factor of $n$.

**Corollary 5.2.** *Under the same assumptions of Theorem 5.1, we give a procedure (see Appendix 4.1) that minimizes GDRO/AFL training error within $\varepsilon$ of R-OPT with probability at least $1 - \delta$ in fewer samples than $\mathcal{O}\left(\frac{D_\Theta C^2 + nC^2 \log(n/\delta)}{\varepsilon^2}\right)$.*

## 6 Empirical Analysis of On-Demand Sampling for Group DRO

This section describes experiments where we adapt our on-demand sampling-based multi-distribution learning algorithm for deep learning applications. In particular, we compare our algorithm against the de-facto standard multi-distribution learning algorithm for deep learning, Group DRO (GDRO) [36]. As GDRO is designed for use with offline-collected datasets, to provide an accurate comparison, we modify our algorithm to work on offline datasets (i.e., with no on-demand sample access).

**Resampling Multi-Distribution Learning (R-MDL).** To adapt our multi-distribution learning algorithm, Algorithm 2, for deep learning applications, we replace its hypothesis-selecting no-regret learning algorithm with a minibatch gradient descent algorithm. We can further adapt our algorithm to offline datasets by simulating on-demand sampling on the empirical distributions of datasets. This modified algorithm, R-MDL, is described in full in Algorithm 5.

In contrast, the GDRO algorithm is also minibatch gradient descent but samples minibatches uniformly from all distributions. Datapoints in each minibatch are importance weighted according to their distribution of origin, where a no-regret algorithm adversarially weights each distribution. Though effective, this GDRO method is brittle and requires tricks like unconventionally strong regularization [36]. Our theory of on-demand sampling suggests that R-MDL should mollify this brittleness.

**Experiment Setting** In Table 2, we replicate the Group DRO experiments of Sagawa et al. [36] and compare the standard GDRO algorithm with our R-MDL algorithm (Algorithm 5). We fine-tune Resnet-50 models (convolutional neural networks) [18] and BERT models (transformer-based network) [11] on the image classification datasets Waterbirds [36, 41] and CelebA [23] and the natural language dataset MultiNLI [42] respectively. We train these models in 3 settings: with standard hyperpameters, under strong weight decay ($\ell$-2) regularization, or under early stopping.

| | | Worst-Group Accuracy | | | Gap in Avg. vs. Worst-Group Acc. | | |
|---|---|---|---|---|---|---|---|
| | | ERM | GDRO | R-MDL | ERM | GDRO | R-MDL |
| Standard Reg. | Waterbirds | 60.0 (1.9) | 76.9 (1.7) | **86.4 (1.4)** | 37.3 (1.9) | 20.5 (1.7) | **8.1 (1.4)** |
| | CelebA | 41.1 (3.7) | 41.7 (3.7) | **88.9 (2.3)** | 53.7 (3.7) | 53 (3.7) | **3.4 (2.3)** |
| | MultiNLI | 66.3 (1.6) | 66.6 (1.6) | **70.3 (1.5)** | 16.2 (1.6) | 15.6 (1.6) | **4.5 (1.5)** |
| Strong Reg. | Waterbirds | 21.3 (1.6) | 84.6 (1.4) | **89.4 (1.2)** | 74.4 (1.6) | 12 (1.4) | **0.4 (1.3)** |
| | CelebA | 37.8 (3.6) | 86.7 (2.5) | **88.8 (2.3)** | 58 (3.6) | 6.8 (2.5) | **1.2 (2.3)** |
| Early Stop | Waterbirds | 6.7 (1.0) | 85.8 (1.4) | **87.1 (1.3)** | 87.1 (1.0) | 7.4 (1.4) | **5.6 (1.3)** |
| | CelebA | 25.0 (3.2) | 88.3 (2.4) | **90.6 (2.2)** | 69.6 (3.2) | 3.5 (2.4) | **0.7 (2.2)** |
| | MultiNLI | 66.0 (1.6) | **77.7 (1.4)** | 43.1 (1.7) | 16.8 (1.6) | **3.7 (1.4)** | 18.3 (1.7) |

Table 2: Worst-group accuracy (our primary performance metric) and the gap between worst-group accuracy and average accuracy, of empirical risk minimization (ERM), Group DRO (GDRO), and our R-MDL algorithm in three experiment settings—standard hyperparameters (Standard Reg.), inflated weight decay regularization (Strong Reg.), and early stopping (Early Stop)—and on three datasets—Waterbirds, CelebA, and MultiNLI. Figures are percentages evaluated on the test split of each dataset, with standard deviation in parentheses. R-MDL consistently outperforms GDRO and performs reliably with or without strong regularization.

**R-MDL consistently outperforms GDRO and ERM.** In every dataset and in almost every setting, R-MDL significantly outperforms GDRO and ERM in worst-group accuracy. In addition, whereas GDRO and ERM have large gaps between worst-group accuracy and average accuracy, R-MDL has almost matching worst-group and average accuracies. This indicates that R-MDL is more effective at prioritizing learning on difficult groups.

**R-MDL is robust to regularization strength.** R-MDL retains high worst-group accuracy even without strong regularization. These results challenge the observation of Sagawa et al. [36] that strong regularization is critical for the performance of Group DRO methods. This suggests that the brittleness of GDRO is due to reweighting rendering the adversary too weak. In contrast, R-MDL provides a robust multi-distribution learning method with significantly less hyperparameter sensitivity.

# 7 Acknowledgments

This work was supported in part by the National Science Foundation under grant CCF-2145898, a C3.AI Digital Transformation Institute grant, and the Mathematical Data Science program of the Office of Naval Research. This work was partially done while Haghtalab and Zhao were visitors at the Simons Institute for the Theory of Computing.

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
