# OpenReview forum: "On-Demand Sampling: Learning Optimally from Multiple Distributions"
_NeurIPS.cc/2022/Conference — NeurIPS 2022 Accept_

### Official Review · Reviewer_XJ3a · 2022-07-11

**Rating:** 7
**Confidence:** 3
**Soundness:** 3 good
**Presentation:** 3 good
**Contribution:** 3 good

**Summary:**

This paper gives improved algorithms for three settings of learning from multiple distributions: collaborative, group distributionally robust, and fair federated learning. Previous results usually bounds the sample complexity by a certain overhead factor multiplied by the number of samples needed for learning a single distribution. In contrast, the sample complexity bounds derived in this work only have *additive* overheads.

In more detail, for $(\epsilon, \delta)$-collaborative learning on hypothesis class $\mathcal{H}$ and $n$ distributions, Theorem 4.1 gives an upper bound of $O(\epsilon^{-2}(\log|\mathcal{H}| + n\log(n/\delta)))$, which exceeds the $O(\epsilon^{-2}\log(|\mathcal{H}|/\delta))$ bound for single-distribution learning by an additive term of $(n\log n)/\epsilon^2$ (when $\delta = \Omega(1)$). In contrast, the previous best bound of Nguyen and Zakynthinou (2018) is at least $(\log n)\cdot(\log|\mathcal{H}|/\epsilon^5)$. Theorem 4.3 gives an improved lower bound, which matches the upper bound when $n$ and $\log|\mathcal{H}|$ are polynomially related.

The approach taken by the authors (which is implicit in prior work) is to view the collaborative learning setting as a zero-sum game between the learner (which selects a hypothesis) and an adversary (which chooses one of the $n$ data distributions). The major task is then to find an (approximate) equilibrium of this game, using as few samples as possible. The key technical observation is an asymmetry between the two players in terms of how much data is needed for getting an unbiased estimate of the gradient, and to what extent these data can be reused for estimating the gradient at different locations.

**Questions:**

1. Are there any natural/simple/structured hypothesis classes for which that the algorithm could be efficiently implemented?

2. The algorithm outputs a (weighted) majority vote of hypotheses. Are previous algorithms also improper, and are there evidence that the problem becomes hard for proper learning?

Minor comments:
- Line 195: What is "VI"?

**Limitations:**

Limitations were not discussed.

**Strengths And Weaknesses:**

This work introduces new tools to the line of research on learning from multiple distributions and further closes the gap between the upper and lower bounds. Both the results and the techniques should be interesting to the community. The new algorithm and its analysis seem highly nontrivial, and I thought the authors did a good job in sketching the proof in the main paper. In summary, this is a solid and well-presented work and should be accepted.

One weakness of the paper is that the algorithms are computationally inefficient. While the main focus is sample complexity, some discussion on the computational aspect seems necessary.

---

> ### Author Response · Authors · 2022-08-01
> **Response to Reviewer XJ3a**
>
> We want to thank the reviewer for their positive feedback and comments. We will address individual comments below.
>
> **Computationally efficient hypothesis classes**:
> We believe that the identification of settings for which computational efficiency can be achieved is an interesting direction for future work. We suspect it may not be generally possible to obtain optimal sample complexity results with computationally efficient algorithms, as our min-max classifier learning problem is closely related to online learning, for which there is a provable computational separation [HK16]. Our algorithms can be efficiently implemented in settings where our no-regret algorithm reduces to optimizing low-dimensional linear functions, which we expect to arise in some regression problems.
>
> **Improper learning**:
> Yes, all previous works used improper learning algorithms. Our algorithms are slightly more general, and allow for either outputting a majority vote of our hypothesis class or a probability distribution over our hypothesis class. In fact, our optimal results are obtained via the latter, which is somewhere between proper learning and improper learning. We do not know if proper learning is hard, although [6] shows some evidence for this, by lower bounding the sample complexity of proper learning under uniform convergence.
>
> **L195 VI:** VI stands for variational inequality. We have revised our manuscript to clarify this initialism.
>
> [HK16]  Elad Hazan and Tomer Koren. The computational power of optimization in online
> learning. In Proceedings of the 48th Annual ACM Symposium on Theory of Computing (STOC), page 128–141, 2016.

---

> > ### Comment · Reviewer_XJ3a · 2022-08-08
> > **Thank you for the reply!**
> >
> > Thank you for answering the questions! I would encourage the authors to include part of the discussion in the final version of the paper. My overall evaluation of the paper remains positive.

---

### Official Review · Reviewer_2WVF · 2022-07-11

**Rating:** 7
**Confidence:** 3
**Soundness:** 3 good
**Presentation:** 3 good
**Contribution:** 3 good

**Summary:**

This paper investigates the on-demand sample complexity of multi-distribution learning. The authors establish the optimal sample complexity of collaborative, group distributionally robust, and agnostic federated learning. These results improve upon the best-known sample complexity in the prior work. The main approach is to regard the problem as a stochastic zero-sum game and apply a variant of stochastic mirror descent approach.

**Questions:**

The paper considers n predefined distributions.
- Can the result generalize to a family of distributions that has a certain bounded capacity, e.g., all exponential distributions?
- Can we reduce the dependence on n if these distributions have some relations, e.g., each one is a uniform distribution on a certain subset of k elements?

**Limitations:**

Not

**Strengths And Weaknesses:**

Strengths:
- The studied problem is important in the machine learning area.
- The theoretical results are optimal, which closes this direction.
- The approaches are standard, which widens the application of stochastic mirror descent.

Weaknesses:
- It may be better to provide some empirical results on the proposed algorithms. The theoretical results will be more convincing if the performance beats the baselines in prior works.

---

> ### Author Response · Authors · 2022-08-01
> **Response to Reviewer 2WVF**
>
> We want to thank the reviewer for their positive feedback and comments. We will address individual comments below.
>
> **Lack of empirical results**
>
> We include a set of experiments in Appendix C.1 where we compare our proposed algorithms against well-known baselines for multi-distribution learning on three datasets. Our algorithms outperform baselines for multi-distribution learning, offering noticeably faster convergence and greater robustness to hyperparameter choices. In the final version of the manuscript, we will summarize these results in the main text.
>
> **Questions**:
> We agree these questions are interesting directions for future work.
>
> Generalizing multi-distribution learning to families of distributions is an intriguing question that could present a novel computationally-efficient method for finding minimax estimators for complicated distribution families. Our presented results can already handle the case where hypothesis risk is convex in the sufficient statistic of a distribution family—this is because we can specify our adversary to optimize over a compact space of sufficient statistics, rather than a probability simplex of mixture distributions. We will include a discussion of this application in the extra page of our final version. We think it would be very interesting to identify other families of distributions for which we could give similar guarantees.
>
> Regarding whether we can reduce dependence on $n$, our lower bound suggests this may be difficult. Our lower bound proves that we cannot reduce dependence on $n$ even if all distributions share the exact same feature distribution and all but one distribution shares the exact same label distribution. On the other extreme, when each distribution constitutes an independent learning problem with its own disjoint support, our sample complexity is bottlenecked by hypothesis class size rather than $n$. However, we agree that identifying distributional assumptions—such as the condition number of our min-max problem---that allow for reduced dependence on $n$ is an interesting question worth pursuing.

---

### Official Review · Reviewer_mxyZ · 2022-07-12

**Rating:** 7
**Confidence:** 3
**Soundness:** 4 excellent
**Presentation:** 3 good
**Contribution:** 4 excellent

**Summary:**

The authors propose learning algorithms for multi-distribution learning problems and establish optimal sample complexity bounds for the proposed algorithms. The authors improve upon the existing results and also correct an error in an existing result to provide lower bounds for their framework. Specifically, the authors present results for two settings of collaborative learning and group DRO formulation. The developed algorithms rely on the stochastic algorithms utilized to solve stochastic zero-sum games.

**Questions:**

This is a theoretical work that improves upon the existing results. The paper is very dense which makes it a little difficult to read, the authors might consider updating the paper by moving some redundant parts to the appendix for improving the readability of the paper.

**Limitations:**

Yes.

**Strengths And Weaknesses:**

Overall, the paper is well written and the presented results are strong. The authors improve upon the state-of-the-art complexity bounds (both upper and lower bounds) by significant factors. The paper feels a bit difficult to read and dense, the authors should try to make the paper more reader-friendly. Here are some comments the authors might consider addressing.

1. The algorithms authors have proposed for solving the problems are based on popular min-max formulations and the algorithms are similar to stochastic gradient descent ascent algorithms. Please comment.

2. It is not clear what is the difference between Algorithm 1 and 2. Both the algorithms look more or less similar. Why both the algorithms are required in the main text of the paper.

3. In lines 198-199 it is not clear why $\text{Err}_v(w^{(1:T)}) \geq \text{Reg-Min}(p,q) + \text{Reg-Max}(p,q)$. Please add a discussion/reference.

4. In eq (7), why the gradients are defined by loss functions. It would be helpful if the authors can clarify.

5. In Line 203 the authors mention with noisy estimates of the gradients the no-regret algorithms become advantageous. Please clarify why this is the case?

6. The authors should discuss the relevance of Lemma 3.1. Moreover, it is not clear why the total cost of randomness is 2t, specifically, why the total cost of randomness is independent of $T$?

7. In Theorem 5.1, why compactness of parameter space $\Theta$ is required? For general optimization problems, this compactness is not required in general.

8. The context of on-demand sampling is not clear from the Algorithms, maybe the authors can add some discussion about the on-demand sampling after stating the algorithms.

#### **Minor Comments:**

- Please define the dimension of $\tilde{w}$ in eq. (4). I assume that $\tilde{w}$ is a constructed by stacking $\tilde{w}_i$'s.
- The use of the term "multi-agent tradeoffs" in the first sentence of the abstract is not clear.
- In the statement of Lemma 2.1 please correct the notation $\\{ w\\}_{t = 1}^T$
- In line 189 $\phi(i,j)$ is not properly defined please clarify that $i \in \mathcal{H}$ and $j \in \mathcal{D}$.
- The definition of the deterministic estimator $h_p^{Maj}$ is not clear.

---

> ### Author Response · Authors · 2022-08-01
> **Response to Reviewer mxyZ**
>
> We want to thank the reviewer for their positive feedback and comments. We will address individual comments below.
> 1. **Relationship with popular min-max formulations**:
> Our algorithm is indeed a stochastic mirror descent/ascent algorithm. What differentiates our algorithm from existing min-max optimization methods is that learning theory tasks do not demonstrate the typical properties of saddle-point problems considered by optimization literature. For instance, our optimization problem does not have smooth gradients nor strongly convex/concave payoff functions. Most importantly, our problem does not involve querying noisy gradient oracles for gradient estimates. Instead, we sample datapoints that can be used to estimate gradients; an important aspect of our problem is studying when we can reuse datapoints for gradient estimation. We have revised our manuscript to include this discussion.
>
> 2. **Clarification of Algorithms 1 and 2**:
> We greatly appreciate your suggestions for making the presentation lighter. Algorithm 1 is an algorithm for finding saddle-points with a datapoint oracle, which we believe may be of independent interest. It is a generalization of Algorithm 2, which is specifically for multi-distribution learning and takes an explicit form. We will revise the paper to lighten the technical presentation of these results.
>
> **Other questions and comments:**
>
> 3. **Inequality on L198-199**: We prove this inequality in Fact B.1, referenced on Line 199. We have revised to make this reference more explicit.
>
> 4. **Why are gradients defined by loss functions**: The payoff function of our zero-sum game is the risk of the learner’s hypothesis mixture on the adversary’s distribution mixture. Both players are optimizing over probability simplexes, so each component of their gradient vectors is the risk of a hypothesis/distribution conditioned on the other player’s distribution/hypothesis mixture. We have revised to include this clarification.
>
> 5. **Why does no-regret help with noisy gradients (L203)**: No-regret algorithms decide a $t$th iterate without looking at the $t$th gradient. This allows one to make useful independence assumptions that linearize noise in online optimization problems. We have revised our current explanation on L204 to clarify this point.
>
> 6. **Relevance of Lemma 3.1**: Lemma 3.1 shows that we can quickly find saddle-points in our noisy zero-sum game. The total cost of randomness should be written as $2T$, not $2t$. We appreciate your catching this typo and have corrected it.
>
> 7. **Why compactness of parameter space**: We, and prior works, assume compactness so that a solution exists for our saddle-point (min-max) optimization problem, as the minimax theorem depends on compactness.
>
> 8. **On-demand sampling**: On-demand sampling algorithms are algorithms that are permitted to iteratively select from which distribution they want to sample a datapoint. We have revised the algorithm section to restate this important definition.

---

> > ### Comment · Reviewer_mxyZ · 2022-08-07
> > **Response to Authors' Comments**
> >
> > I thank the authors for clarifying my comments. I will keep my score. Thanks!

---

### Official Review · Reviewer_pzrG · 2022-07-20

**Rating:** 8
**Confidence:** 3
**Soundness:** 4 excellent
**Presentation:** 3 good
**Contribution:** 4 excellent

**Summary:**

This paper studies the problem of multi-distribution learning. Specifically, the paper proposes several sample complexity bounds for on-demand learning. The central contribution of the paper is to first frame collaborative learning as a zero-sum game with stochastic payoff and then use tools from the framework of stochastic mirror descent to obtain sharp rates for various collaborative learning settings. The authors obtain near-optimal rates for agnostic collaborative learning, group DRO, and agnostic federated learning.



**Questions:**

- I am curious to know if it is possible to obtain data-dependent bounds for the 3 settings under this framework?

**Limitations:**

The authors have not explicitly discussed limitations of their approach. Since this work is theoretical, it is unlikely it will have potential negative societal impacts.

**Strengths And Weaknesses:**

+ The paper is well organized and very easy to understand. The main contributions are listed clearly, and the authors provide a good overview of the approach as well.
+ The approach appears to be novel and is a general technique to handle learning from multiple distributions. The authors provide near-optimal rates for a variety of problems that improve over the prior state-of-the-art by a significant amount.
+ The experimental results confirm an improvement in worst-case accuracy on 3 datasets. It would be beneficial to see performance on a dataset with larger variation than WaterBirds and Celeb-A (e.g., YFCC), but that is beyond the scope of this paper.

---

> ### Author Response · Authors · 2022-08-01
> **Response to Reviewer pzrG**
>
> We want to thank the reviewer for their positive feedback and comments.
>
> We agree that the question of data-dependent bounds for multi-distribution learning is interesting. The sample complexity of multi-distribution learning depends not only on how quickly each data distribution concentrates but also the condition number of our min-max problem. However, traditional data-dependent Rademacher complexity analysis only captures the former. We think that a very interesting direction for future research is to identify data-dependent bounds for on-demand sampling that are capable of addressing both of these aspects.

---

> > ### Comment · Reviewer_pzrG · 2022-08-03
> > **Thank you for the response**
> >
> > Thanks authors for responding to my question. I'll maintain my score of 8.

---

### Meta-Review · Area_Chair_fNdZ · 2022-08-20

**Recommendation:** Accept
**Confidence:** Certain

**Metareview:**

This paper studies multi-distribution learning; it formulates the problem as a zero-sum game with the stochastic payoff and then uses tools from the framework of stochastic mirror descent to obtain optimal sample complexities for various collaborative learning settings.  All reviewers are very positive about this paper: interesting problems, nice techniques, and optimal results.

**Award:**

Yes

---

### Decision · Program_Chairs · 2022-09-14

Accept